# Optimization Strategy for Process Design in Rubber Injection Molding: A Simulation-Based Approach Allowing for the Prediction of Mechanical Properties of Vulcanizates

**DOI:** 10.3390/polym16142033

**Published:** 2024-07-17

**Authors:** Martin Traintinger, Maurício Azevedo, Roman Christopher Kerschbaumer, Bernhard Lechner, Thomas Lucyshyn

**Affiliations:** 1Polymer Competence Center Leoben GmbH, Sauraugasse 1, 8700 Leoben, Austria; martin.traintinger@pccl.at (M.T.);; 2Department of Polymer Engineering and Science, Chair of Polymer Processing, Montanuniversitaet Leoben, Franz-Josef Straße 18, 8700 Leoben, Austria; thomas.lucyshyn@unileoben.ac.at

**Keywords:** injection molding, rubber part quality, simulation, mechanical characterization, logistic growth, optimization, sustainability

## Abstract

Selecting the optimal settings for the production of rubber goods can be a very time-consuming and resource-intensive process. A promising method for optimizing rubber processing in a short period of time is the use of simulation routines. However, process simulations have only recently enabled meaningful predictions of not only the part’s state of cure but also its mechanical characteristics. As a first approach, second-order polynomials were considered suitable for describing the properties of compression-molded parts. However, more precision is required for injection molding due to the narrower distribution of mechanical characteristics of parts produced at different vulcanization temperatures. This became evident when the approximation of mechanical data with second order models partly revealed significant failures of part behavior prediction. To tackle this issue, a combined approach for approximation is proposed in this contribution by means of logistic growth function in addition to second order polynomials. To feed the model, an experimental plan was designed for producing injection-molded parts from an SBR compound at various temperatures and to different degrees of cure. The parts obtained were then characterized mechanically, and the results were opposed to varying degrees of cure and extents of reaction to calculate the model coefficients. Once available, a simulation-based calculation of the mechanical part quality is possible. The comparison of test results from the simulation and the real process has shown a reliable prediction, as simulation results were found within the natural deviation of the real measurements.

## 1. Introduction

At a time when the relation between waste management and climate change is discussed almost daily, it is clear that scientific research into ways to improve sustainability is crucial to tackling one of the key challenges of our modern age. In the field of rubber processing, this becomes even more critical due to the irreversibility of the curing reaction that makes rubbers insoluble, thermally stable and, hence, difficult to recycle [1]. Basically, there are ways to give vulcanizates a second life, e.g., on sports grounds, in concrete or in road construction, but also after so-called devulcanization, in which the crosslinks are selectively disrupted by a treatment, e.g., chemically, thermomechanically or by irradiation, and the polymers are reintegrated into another compound [2]. However, none of these methods completely solve the sustainability issue, as either the toxicity of the chemicals or the energy consumption plays a role, which only shifts the challenge to another area. The most efficient way to prevent waste in the rubber industry is therefore to optimize the manufacturing process such that waste production is limited to an absolute minimum. A truly old-fashioned but still popular method of optimizing a manufacturing process is to test a new mold system or rubber compound by trial and error experiments [3]. This involves adjusting various variables, usually machine settings such as injection volume rate or vulcanization time, until the molded part meets certain requirements. The main disadvantage, though, is that it can be very time-consuming and resource-intensive, depending on how quickly the ideal settings for a system of material, mold and machine are found.

More efficient ways of optimizing processes and adapting the production chain with regard to zero-waste production and quality enhancement were part of various research projects. Berkemeier et al. [4,5] proposed a dynamic approach of controlling rubber injection molding in which the vulcanization time is adjusted based on an energy balance calculated from the process steps. Hutterer et al. [6], on the other hand, developed an attempt based on principal component analysis to assess whether parts produced are of good quality. In the thesis of Ryzko [7], process control is derived from a statistical analysis assessing the obtained part quality as a function of different machine settings. A similar approach was elaborated by Traintinger [8], though measurable process signals, e.g., injection pressure or work conducted during injection, were correlated with the final part quality. Despite the range of different possibilities for process optimization, each of the methods mentioned before is to some extent dependent on practical experiments carried out in advance, which in turn can lead to a vast amount of rubber waste.

Another approach is the use of simulation software to optimize manufacturing in rubber processing. Fasching [9] pursued this idea in his thesis by establishing a correlation between process signals and the quality criteria of cured rubber parts. Until recently, the degree of vulcanization, derived from normalizing kinetic data, was apparently the only quality criterion available in the simulation. This value is used to evaluate the final quality, assuming that maintaining the degree of cure is crucial. In reality, parts reveal significant mechanical differences when vulcanized to the same degree of cure at various temperatures. This was demonstrated in the work of Hornbachner [10] by producing elastomer parts with a target state of cure of 80% at temperatures between 140 °C and 170 °C, revealing a maximum difference of 36% in compression set within the studied temperature range. Traintinger et al. [11] confirmed these results when performing a multi-stage swelling analysis, including a chemical treatment of cured rubber samples providing information on the poly-, di- and monosulfidic crosslinks. These results prompted Weinhold et al. [12] to develop a new approach that considers the materials process history, and which allows for the prediction of mechanical part behavior from simulations.

In this contribution, the intent was to prove the applicability of Weinhold’s approach in simulation for injection-molded rubber parts. For this purpose, a systematic design of the experiment was set up with the aim to provide the proposed model with data from the compound’s reaction kinetics, and with quality data from mechanical testing. The latter have been measured via dynamic mechanical analysis in compression mode and determination of the dynamic spring constant, as well as by conducting compression set analysis. In the first round, though, it was found that the data range given by the experimental design was not fitted entirely with a second order model. Therefore, an alternative model was investigated by combining the second order model with a logistic growth function. Following the approximation of the model coefficients, various simulation runs were started, each of which was set up according to real experiments conducted in the test facility. The results, mechanical properties derived from simulation on the one hand, and the properties measured from injection-molded parts on the other hand, have been opposed to each other, aiming to validate the model proposed in this research.

## 2. Theoretical Background

Regarding the thermal history of a rubber part during processing, especially inside the cavity, it is rather logical that temperature does not exhibit constant levels. Instead, the temperature evolves locally, resulting in a distinct difference in the state of cure, which emphasizes the decisive roles of not only the vulcanization temperature but also the time on the curing reaction taking place. Supported by simulation programs and numerical modeling, it is possible to conduct element-wise calculations on the evolving state of cure, which could be translated to related information on common part characteristics such as compression set or dynamic mechanical behavior. The calculation of mechanical part characteristics within the simulation software was not available until recently, and it was eventually only approached by further treatment of simulation results and calculations in an external program. Therefore, the herein presented approaches of determining these physical figures mark a novelty in the simulation routine, which is not yet state of the art.

The first steps in developing a suitable method for the aforementioned purpose in simulation were realized by Weinhold et al. [12,13,14] by proposing an approach considering the temperature-affected average curing speed to predict mechanical properties *Y* of compression-molded parts. Their calculation is based on the choice of a second order polynomial given in Equation (1), as follows:(1)$$Y=K+\alpha_{1}c+\alpha_{2}c^{2}+\beta_{1}\langle \dot{c}\rangle +\beta_{2}{\langle \dot{c}\rangle }^{2}$$
where *K*, *α*_1,2_ and *β*_1,2_ are model coefficients and *c* and <*ċ*> represent the degree of cure and the average curing speed, respectively. The degree of cure *c* at any time *t* is calculated from Equation (2), as follows:(2)$$c=\frac{S_{t}-S_{0}}{S_{M}{-S}_{0}}\cdot 100$$
where the minimal torque *S*_0_, the maximal torque *S_M_* and the transmitted torque at any time *S_t_* are obtained from measurements of the curing kinetics.

Weinhold’s approach for the average curing speed $\langle \dot{c}\rangle$ is then introduced by Equation (3), as follows:(3)$$\langle \dot{c}\rangle =\frac{1}{t_{M}}\int_{t=0}^{t=t}\dot{c}(T)\;dt$$
where *ċ* is the current curing speed and *t_M_* is the vulcanization time at maximal torque. Locally resolved variables are then introduced to the model, giving a locally resolved response of mechanical part behavior in return.

From a mathematical viewpoint, however, second order polynomials do not necessarily evolve enough accuracy to properly fit sigmoidal-shaped curves like those obtained from kinetic measurements. Fasching [9] addressed this issue in the context of the design of the experiment together with the contributing space of the experiment, and they mentioned that insufficient approximation quality was obtained if, e.g., incubation time and time close to the completion of the reaction were considered. In that case, he suggested the introduction of a logistic growth function and proposed a combination of first order model and logistic growth term for his purpose. Nonetheless, it should be emphasized that the ability of Weinhold’s second order model to predict mechanical part properties by simulation was approved for compression-molded parts, meaning that the method should not be categorically excluded. However, initial tryouts have revealed that this approach does not work properly for parts being injection molded from the same rubber compound, which could be due to shear-induced effects affecting polymer morphology and filler orientation.

Logistic growth functions are capable of approximating sigmoidal curves, which appear from kinetic measurements in rubber processing. Amongst various types of nonlinear mathematical models describing time-dependent growth *Y*(*t*) is the Richard’s function [15], also referred to as a generalized logistic model [16,17], given in Equation (4):(4)Yt=F(1+exp⁡−γt−δ)1ε
where *γ* is the growth rate of the exponential phase, *δ* represents the turning point and *ε* is a factor corresponding to the shape of the curve. The forth coefficient, *F*, is the upper asymptote or the carrying capacity, which describes the upper limit of the function that is not exceeded. In terms of mechanical characteristics of cured rubber parts, *F* could represent, for example, the best value achievable for an arbitrary compound. Assuming the optimum dynamic spring constant of a rubber part being fully cured is 500 N mm^−1^, the upper asymptote would take the value 500, or at least a value very close to it.

For non-isothermal processes like injection molding, however, the material experiences various stages of increasing temperature, which affects the mechanical properties of the part until the end of the cycle. Hornbachner [10] investigated the mechanical part characteristics of compression-molded parts, which were manufactured to the same degree of cure *c* at various vulcanization temperatures. She showed that parts reveal higher values of the dynamic spring constant *C_DYN_* when cured at 140 °C to a degree of cure, e.g., *c* = 80%, compared to parts vulcanized at, e.g., 170 °C to the same degree of cure. In Traintinger et al. [11], this behavior was investigated in more detail, and the results on mechanical behavior were extended by chemical analysis. In a comparison of parts cured at 160 °C and 170 °C, respectively, it was shown that the amount of mono- and disulfidic crosslinks increases with temperature. In reverse, this means that more content of polysulfidic bridges between the polymer chains is found at the lower of the two temperatures. With respect to the dynamic behavior, this means that parts become stiffer with rising temperatures. In order to contribute to these results, we concluded that a constant value for the upper asymptote *F* is insufficient, as it may not consider non-isothermal conditions. Instead, the herein proposed approach should be capable of predicting the entire trend of mechanical behavior, e.g., the dynamic spring constant, appropriately. Speaking of parts cured to the same degree of cure at various temperatures, this means that the approach must be able to realize that higher values of dynamic spring constant are to be expected at lower temperatures, which is indicated in Figure 1 with parts cured to *c* = 50% and *c* = 90% at four temperatures ranging from 140 °C to 170 °C. By applying a second order polynomial, it was found that the measured dynamic behavior for parts cured at 140 °C is predicted suitably in either case. For this reason, the second order equation was defined as the upper asymptote.

However, Figure 1a also reveals that an allone-standing approximation of the dynamic behavior via second order polynomial is not enough to contribute to the temperature-dependent trend of dynamic behavior obtained at constant degrees of cure. While appearing appropriate for parts cured to *c* = 90%, it is clearly indicated that the prediction becomes inadequate for parts cured to *c* = 50% at temperatures above the aforementioned. For this reason, and in order to contribute to the temperature effect obtained from the curing reaction, a combination of Weinhold’s second order model, given in Equation (1), and the generalized logistic model, stated in Equation (4), both modified for the herein addressed purpose, is introduced by Equation (5):(5)Y(t)=K+α1c+α2c2+β1X+β2X2(1+exp⁡−γX−δ)1ε

In this equation, the Greek letters *α*–*ε* and *K* mark model coefficients, which are determined from the reference data matrix containing test results obtained from part characterization, and the degree of cure *c* is calculated with the previously mentioned Equation (2) from kinetic data, while the extent of reaction *X* is calculated as integral from the normalized vulcanization isotherm in the range between the incubation time *t_i_* and any time of an arbitrary degree of cure *t* via Equation (6).
(6)$$X(T)=\int_{t_{i}}^{t}c(T)\;dt$$

As indicated in Figure 1b, the approximation of measured part characteristics is significantly improved when applying the combined approach stated in Equation (5). Different to the previous case of using the second order model, the adapted method also captures the dynamic behavior of parts cured at higher temperatures. The benefit resulting from the application of this approach should become apparent in the simulative prediction of component properties. In the subsequent sections, the proposed approach’s suitability in Equation (5) is validated, focusing on the question whether the additional fit parameters evolve higher prediction accuracy and if the nonlinearity of the test results could be approximated more accurately.

## 3. Materials and Methods

### 3.1. Materials

Aiming to study the proposed new model approach, a styrene-butadiene rubber compound (SBR) provided by Semperit Technische Produkte Gesellschaft m.b.H. (Wimpassing, Austria) was employed. The highly filled compound contains carbon black and white fillers exhibiting a ShoreA hardness of 70 and a sulfur-based curing system. Detailed quantities of the compound’s ingredients cannot be disclosed due to its application for industrial purposes.

### 3.2. Material Characterization

Analyses of the SBR’s reaction kinetics and the determination of the curing behavior were conducted by isothermal oscillatory shear experiments according to 1983:03 DIN 53529-1 and DIN 53529-2 [18,19], employing a rubber process analyser (RPA) from Montech Werkstoffpruefmaschinen GmbH (Buchen, Germany) at the following different temperatures: 140 °C, 150 °C, 160 °C and 170 °C. While these settings contribute to the approximation model that is later on applied to simulate part characteristics, additional measurements were conducted for evaluation purposes at 155 °C and 165 °C. Deformation and frequency were kept constant at 0.5° and 1.667 Hz, respectively. The measurements were terminated automatically as soon as the testing device recorded a plateau of the transmitted torque. Subsequently, the data was normalized in the range of minimum and maximum torque, providing vulcanization isotherms utilized to derive the vulcanization times for the experimental work. Additionally, the material data sets for virtual validation experiments are fed with the same curves and are approximated with suitable models. In this contribution, the Kamal and Sourour model [20] was employed for this purpose. The model coefficients are provided in the appendix, along with further information on the rheological and thermal properties.

Apart from the transmitted torque, the reaction rate was also obtained during the RPA measurements. Those recordings were further processed to calculate the extent of reaction in the range between incubation time and any arbitrary time up to an extent of 100%. This contributes to the original Weinhold approach, considering the curing reaction only and dismissing especially the effect of reversion. The extent of reaction is, besides the degree of cure, the second input variable necessary to feed the model function aiming to calculate mechanical characteristics of cured rubber parts.

### 3.3. Injection Molding

Determining part characteristics virtually first requires access to appropriate reference data, which provides the model with the information necessary to approximate its coefficients. Therefore, rubber parts were produced on a rubber injection molding machine MTF750/160editionS from Maplan GmbH (Kottingbrunn, Austria). The machine was equipped with a multi-cavity-mold shown in Figure 2, including a cold runner system from PETA Formenbau GmbH (Bad Soden-Salmuenster, Germany).

Pressure and temperature (pT)-sensors from Kistler Instrumente AG (Winterthur, Switzerland) were mounted to record processing conditions from cycle to cycle. At four temperatures, 140 °C, 150 °C, 160 °C, and 170 °C, parts were produced from the SBR compound. The vulcanization times were derived from the corresponding isotherms. At each temperature, the same degrees of cure *c* were targeted, whereby five different levels were deployed evenly distributed in the range of *c* = 50–90%. All experimental settings were conducted in quintuplicate, allowing for statistical validity in further investigations. A summary on the settings applied in the reference phase is given in Table 1. It should be pointed out explicitly that the mentioned cooling phase was implemented in order to preserve the degree of cure and to prevent any post-curing reactions after ejection as much as possible. For this reason, parts were placed in an ice-cooled bucket of water at least for the time given.

The applicability of the new model for predicting part behavior was investigated using production data from the processing of the SBR compound. From the same batch being deployed in the reference phase, additional parts were produced for the validation according to the following requirements: (i) the vulcanization temperature should be embedded within the range of the reference data, but it should not equal an already existing condition, and (ii) the degree of cure should be within the given limitations as well. Therefore, at 155 °C parts were cured to the corresponding vulcanization time for c = 75%, while at 165 °C the studied degrees of cure were c = 60% and 75%. A list of the entire settings of the validation experiments is given in Table 2.

### 3.4. Simulation

For the simulation of the aforementioned validation experiments on injection-molded rubber parts, *SIGMASOFT^®^* Version 6.0 (SIGMA Engineering GmbH, Aachen, Germany) was employed. Therefore, the digital twin of the manufacturing process was set up according to the experimental outline of the reality, including appropriate definitions of (i) the material data set, (ii) the mold, (iii) boundary conditions, and (iv) process settings referring to the phases dosing, injection and vulcanization, which are summarized in Table 3. For our investigations, the compound has been characterized kinetically, rheologically and thermally, and the data were processed accordingly. Details on the data curves and the approximation models are presented in the appendix. 

The 3D tetra mesh, which is highly affecting duration and qualitative outcome of a simulation, was defined individually for the part and the mold. A suggestion by the software supplier asks for a minimum of three cubic mesh cells anywhere along the material-containing elements. In our case, a die with a diameter of 2 mm was applied in the mold, hence the size of the equidistant cubic mesh cells was chosen to result in five elements in plane direction at this position. Consequently, the part is then exhibiting a total of 14 mesh elements in the direction of thickness, as shown in Figure A4a, sketching a section view of the meshed part.

Amongst the considerable boundaries in setting up the simulation is the choice of mold material. In our calculations, steel of the type 1.2312 was used as mold material for those sections exhibiting material flow channels or cavities. For all other parts, such as the mold’s clamping plates or the screws, steel of the type 1.1730 was applied. By choosing these types of steel we contribute to the real mold used for the practical experiments. Furthermore, heat transfer coefficients (HTC) mark another sort of boundary condition that could be adapted individually. Herein, however, the software-suggested default values were considered in the calculation, which were 10 kW m^−2^ K^−1^ for steel–steel contact surfaces and 0.8 kW m^−2^ K^−1^ for part-steel surfaces, respectively.

Ultimately, the process settings are defined according to the real process. It should be emphasized that the consideration of the dosing phase follows an approach published by Traintinger et al. [21] which relies on a rather unique test stand developed by Kerschbaumer [22]. This device, which is designed according to the dosing unit of the injection molding machine being deployed in our experiments, enables the observation of the actual state of the mass temperature after dosing, which depends on the adjustable cylinder wall temperature, the screw speed and the back pressure. In Traintinger et al. [21], it was stated that higher accuracy of the developing thermal history of the material can be expected in the virtual process if the time-dependent state of temperature is considered accordingly, hence having an expectable effect on the ultimate part quality as well. Therefore, a representative temperature profile valid for the SBR compound, depicted in Figure A4, was implemented in the simulation experiments. 

### 3.5. Part Characterization

Regarding the assessment of the injection-molded rubber samples, mechanical testing methods were employed. In the simulation, evaluation areas are considered to mark the same locations of the characterization.

#### 3.5.1. Mechanical Testing

Mechanical testing of the rubber parts started with a non-destructive dynamic mechanical analysis (DMA) employing an ElectroPuls^TM^E3000 from Instron GmbH (Norwood, MA, USA). A displacement-driven test run developed by Hutterer et al. [6] was applied to determine the dynamic spring constant *C_DYN_*, which was used for further comparisons. Figure 3a indicates the testing location for DMA. 

In addition to non-destructive DMA, a compression set (CS) analysis was conducted to observe the remaining deformation of a cured rubber part after it has been compressed and stored in an oven. Following the standards given by DIN-ISO 815-1 [23], samples with a diameter of 13 mm were compressed by 25% and stored at 70 °C for 24 h prior to the measurement of the remaining deformation. The specimen’s location within the injection-molded part is sketched in Figure 3b.

#### 3.5.2. Evaluation Area

In the simulation, any physical change evolves locally, thus visualizing, e.g., the temperature or the state of cure from the inlet until the very last corner in the cavity. For quick estimations, so-called picked points could be attached in *SIGMASOFT^®^*, displaying any result desired at the chosen point. However, this single-point-value only represents the physical state from the related cubic mesh cell and does not consider the conditions met in the neighboring cells, neither in the same plane nor in depth. Therefore, the response provides very little information, depending on the defined mesh settings. 

However, more informative figures are observed when so-called evaluation areas are introduced. These areas mark a domain region in the part which is used solely for calculation purposes, hence it affects neither the mesh generation nor the simulation itself. Depending on the dimensions, which could be defined individually and according to the desires, all mesh elements covered by the area are considered, and the result, such as the degree of cure, represents the mean value of all elements. Location and size of the evaluation areas are defined according to the positions of part characterization. Evaluation areas with a diameter of 13 mm and a height of 6.3 mm were defined to mark a CS sample, and 1.26 mm thick areas with a diameter of 8 mm were considered for comparing the simulation results to test results from DMA. It should be emphasized explicitly that the thickness of the DMA-related evaluation area was chosen according to the setup met in the real tests. There, a probe conducts a 20% compression from the surface prior to relaxation and cyclic loading. For this reason, we herein assume that only the upper 20% of the vulcanized rubber part is mostly relevant for the comparison to the simulation results. However, it is important to state that, due to the sample’s viscoelastic nature and the stress propagation, the real compressed region is larger than the one limited by 20% of the thickness. Furthermore, it is very likely that the evaluation region is not perfectly cylindrical, which imposes further complexity to the analysis that will not be covered in this study.

## 4. Results

Predicting part characteristics in an injection molding simulation on behalf of the herein introduced generalized logistic model (Equation (5)) requires access to kinetic data from the raw rubber compound. The data, obtained as transmitted torque *S* from RPA measurements, indicates the temperature-dependent curing reaction and provides information of the function variables degree of cure *c* and extent of reaction *X*. Figure 4a–c depicts transmitted torque *S*, degree of cure *c* and the extent of reaction *X* of the SBR compound applied for generating the reference data.

In a former work [11], the effect of vulcanization temperature on part characteristics was investigated for compression-molded parts cured to the same degree of cure. The outcome was a significant difference in the crosslink density and in the balance between mono-, di- and polysulfidic crosslinks. At a temperature of 160 °C, higher crosslink density was obtained, which was suggested to be mainly a result of the longer vulcanization time, while at 170 °C, the amount of mono- and disulfidic bonds rises, caused by the elevated energy input [11]. The same conclusion can be drawn from the measured transmitted torque as well, where the curves exhibit higher maxima when the vulcanization temperature is reduced, suggesting that a more pronounced state of cure can be obtained with lower temperatures. In case of the sulfur-cured SBR being characterized, the difference of torque between the samples cured to 140 °C and 170 °C is *ΔS* = 3.0 dNm, which is roughly 20% (Figure 4a). For deriving the vulcanization time required to obtain a defined degree of cure *c* and for process simulations, the raw data are then normalized by applying Equation (2) to obtain *c* (Figure 4b); however, through this, decisive information on the mechanical behavior is lost and the isotherms no longer provide information on the expectable mechanical difference caused by various vulcanization temperatures [10]. This impairs common simulation routines to the extent that results relating to the curing reaction are only displayed as the state of cure. An integrative simulation, which considers the history of the processed compound, is thus omitted and the calculation of the mechanical behavior of the part requires additional programs. With the proposed approach, prediction of the injection-molded parts’ mechanical properties is realized within the simulation software. Therefore, the extent of reaction *X* is suggested as an additional variable besides the degree of cure. 

Applying Equation (6), the extent is calculated from the degree of cure at each temperature considered in the experiment, which results in the data curves indicated by Figure 4c and listed in Table 4. This again reveals the expected difference in mechanical behavior. As with the transmitted torque, it can be concluded that parts cured at 140 °C exhibit, e.g., more favorable dynamic behavior due to the occurrence of longer sulfidic crosslinks compared to parts cured at an above situated temperature [11].

For the validation experiments showing the benefits of applying the herein proposed approach of calculating mechanical part behavior on behalf of process simulations, the SBR compound’s curing kinetics under study are depicted in Figure 5a–c. The graphs reveal additional kinetic data measured at 155 °C and 165 °C, which represent temperatures within the space of the experiment that were not considered for the model calculation.

Aiming to provide the logistic model with appropriate information on the mechanical behavior of rubber goods, parts were produced upon injection molding according to the previously described methods. The parts were characterized with common methods, leading to the dynamic spring constant *C_DYN_* obtained from DMA measurements and the compression set *CS* in Figure 6a,b as a function of time and temperature. As in our previous work [11], parts were cured to the same degree of cure at various temperatures, confirming that quality maintenance is not possible when parts are produced under these circumstances. Comparing, e.g., the results from the dynamical measurements, the spring constant, observed from parts cured to *c* = 50% at 140 °C and 170 °C, varies by approximately 20%, whereby the results at the lower temperature were interpreted as more favorable simply due to the fact that it approaches higher levels. With increasing degrees of cure, the ratio between the same temperature range is reduced, but a 13% variation is still observed at *c* = 90%, which represents the highest degree of cure considered in this experiment. Regarding the compression set, an even greater ratio was obtained from the results, being 35% for parts cured to *c* = 50% and 20% at the highest degree of cure for the same temperatures. In our recently published work [11], the issue of declining ratios has been addressed when comparing parts cured at different temperatures and an increasing degree of cure. Regarding the dynamic mechanical analysis, it was found that samples cured to the same degree of cure at various temperatures display higher stresses when cured at lower vulcanization temperatures. This appearance was related to the effect of time, which apparently has more impact on the curing reaction than the temperature or the energy being put into the part, especially if these were cured to a degree of cure less than 80%. It was concluded that more crosslinks have been formed when parts were cured to, e.g., *c* = 80% at 160 °C compared to parts cured to the same degree of cure at 170 °C. However, it was also argued that the amount of energy being put into the reaction, which is obviously higher at elevated temperatures, can nearly make up the difference, leading to the declining ratio of part characteristics of rubber goods cured at a degree of cure above 80% at different temperatures. These findings represent another emphasizing argument to develop suitable approaches for part behavior prediction.

In order to obtain access to the desired mathematical function capable of predicting mechanical part characteristics when applied in simulation, the experimental results are plotted according to Figure 7, showing the mechanical test result over the extent of reaction and in dependence of the degree of cure. This follows the originally suggested path introduced by Weinhold et al. [12], except for one point: instead of the average curing speed calculated from Equation (3), the curing reaction’s extent is employed, which was found to be more suitable due to resolution issues. An initial attempt correlating the mechanical properties of the cured parts with the average curing speed has revealed that insufficient resolution is given between the test results. Consequently, the model could not clearly distinguish, e.g., whether the parts were cured to *c* = 70% or to *c* = 90%. We assume that this is due to additional shear-induced effects associated with injection molding, such as disentanglement of physically entangled macromolecules and change in the filler structure’s morphology, resulting in narrower test results for parts cured at different temperatures. This is in contrast to the method of compression molding, where the material being processed is exposed to fewer shear effects, as no dosing or injection step is required. In addition, the average curing speed is calculated as a slope between two points, which causes an irregular error. This fact in combination with the narrow distribution of part properties could be the reason why the first approach with the average curing speed was inappropriate. To optimize the approach, the exact area between the same two points was instead considered for the extent of the reaction, resulting in a more reliable and less erroneous estimate. In the graphs plotted in Figure 7, it is observed that the dotted lines, indicating the approximation of the measured mechanical behavior, follow their chronological order, thus implying the ability of distinguishing the properties being the result of various vulcanization time steps. The fit coefficients derived from the experimental data are listed in Table 5.

Another point that should be discussed at this stage is the approximation on behalf of logistic growth functions, and, in those terms, the benefit of such functions over the use of second order polynomials, which were originally suggested by Weinhold. Bearing in mind the natural progress of a curing curve of an ideal rubber compound, it occurs that the shape of the curve is sigmoidal, which is seen, e.g., in the curves of transmitted torque in Figure 4. Similar observations are expected when comparing mechanical properties of injection-molded rubber parts as a function of time at a given temperature. Thinking now of a second order model using, e.g., the least square method to fit the real data and calculate the function coefficients, it appears that a certain range of the known data is fitted well, while the fit starts to increasingly deviate from reality near the edges [9]. This is trivial mathematical knowledge. For the experiments being presented, it was found that the second order polynomial was able to fit the mechanical behavior of parts cured to *c* = 80% and *c* = 90% properly, but it revealed a lack of predictability of properties being dedicated to a lower degree of cure. This issue is indicated in Figure 1a, which compares parts cured to *c* = 50% and *c* = 90%. Taking, for example, parts cured to *c* = 50% at 170 °C, a dynamic spring constant around *C_DYN_* = 420 N mm^−1^ was observed where the model expected a value of *C_DYN_* = 465 N mm^−1^. However, comparing the same constellation with the logistic growth function, the model’s prediction value was found to be *C_DYN_* = 422 N mm^−1^, which is obviously in favor for the purpose of correct part behavior prediction. From another view, we observed that the approximation had no benefits for parts cured to *c* = 80% and above if the logistic fit is applied instead of the second order approach. Indeed, both methods revealed the same predictions for those parts. The benefit of the logistic model, though, is that the mechanical behavior of parts cured to lower degree of cure, e.g., *c* = 50%, predicts the real characteristics of these parts more accurately, as described before. One reason for the observed accordance is the factor *ε* in Equation (5), which is a shape factor that allows precision tuning of the approximation. Assuming an extreme scenario with a rather small shape factor, e.g., *ε* < 1, it appears that the fit function for parts cured to a degree of cure below *c* = 70% becomes relatively steep, tending to approach a vertical line the smaller the factor gets. Opposed to that, assuming *ε* is in the range of 1000, the shape factor becomes less effective and the approximation tends to describe the same trend as was observed from a second order polynomial. Finding the most suitable coefficients to approximate the mechanical behavior can be achieved by employing any solver available, e.g., by using Matlab’s lsq-curvefit-function, which is supposed to solve nonlinear least square problems. As soon as access to the function is established, the information can be transferred to the simulation software to determine mechanical part characteristics from data obtained in the virtual injection molding process.

For validation of the proposed method of predicting mechanical part characteristics via simulation, parts were produced upon injection molding at process settings that were within the defined space of the experiment but which were not considered in the model approximation. Following that, the rubber parts were mechanically characterized with dynamic mechanical analysis and a compression set. The results are depicted in Figure 8, together with the predictive value obtained from simulation. In addition, they are summarized in Table 6. For the results of parts produced at *T* = 155 °C and *T* = 165 °C, it is observed that the calculated mechanical behavior from simulation is found well within the natural deviation of the measurements. Consequently, it can be concluded that the logistic growth function is capable of describing the reality if the degree of cure and the extent of reaction obtained in the simulation are placed properly in the fit function.

## 5. Conclusions

Optimizing the process in rubber manufacturing can be a rather sophisticated and time-consumptive endeavor. Quite frequently, the strategy follows a simple trial and error method where only one factor is varied at a time until the molded parts meet the designated specifications. As a result, a vast amount of resources are wasted, which should be avoided due to environmental and sustainability concerns and cost aspects.

In this contribution, a new approach was introduced, allowing for a sustainable optimization of the process setup in rubber injection molding. The aim was to establish an integrative simulation routine that enables precise predictions of mechanical part characteristics under consideration of the processing history of the rubber compound. By employing a combined model of logistic growth and second order polynomial, practical test results, e.g., the mechanical behavior obtained from dynamic mechanical analysis or compression set can be described mathematically, if they were related to the degree of cure and the extent of reaction. It has been shown that a rather simple design of experiment, being based on a variation in vulcanization time and vulcanization temperature, delivers sufficient data to feed the mathematical model. Subsequently, employing the model on injection molding simulations delivered unambiguous evidence that mechanical part properties observed from real experiments can be predicted well within the natural deviation of such measurements. Therefore, we are convinced that this method has high potential to affect optimization strategies in industrial scale manufacturing positively in a sustainable and resource-saving way. However, there is still space for further experiments. At the current stage, the methodology was elaborated intensely for one compound only, using sulfur as a curing agent. We admit that the herein presented way might need adjustments for other compounds with, e.g., peroxides as the curing system. Also, compounds revealing marching modulus or reversion behavior in the curing kinetics may require different approaches. This will be investigated in detail in subsequent research.

## Figures and Tables

**Figure 1 polymers-16-02033-f001:**
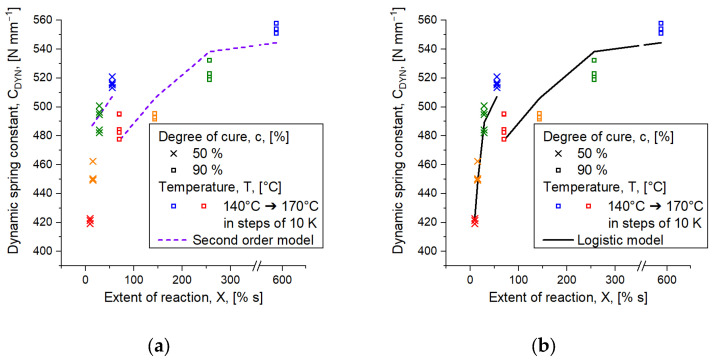
Comparison of approximating dynamic part characteristics with various mathematical approaches. Parts cured to a degree of cure *c* = 50% are marked with crosses, whereas parts cured to *c* 90% are indicated by squares. The colors represent the vulcanization temperatures 140 °C (blue), 150 °C (green), 160 °C (orange), and 170 °C (red). (**a**) Indicates the trend of second order polynomial as an approximation model. While the behavior obtained from parts cured at 140 °C is predicted adequately, the opposite is the case at higher temperatures and for parts cured to *c* = 50%. (**b**) Data approximation with the combined logistic growth and second order polynomial as upper asymptote fits the entire trend of dynamic part behavior well.

**Figure 2 polymers-16-02033-f002:**
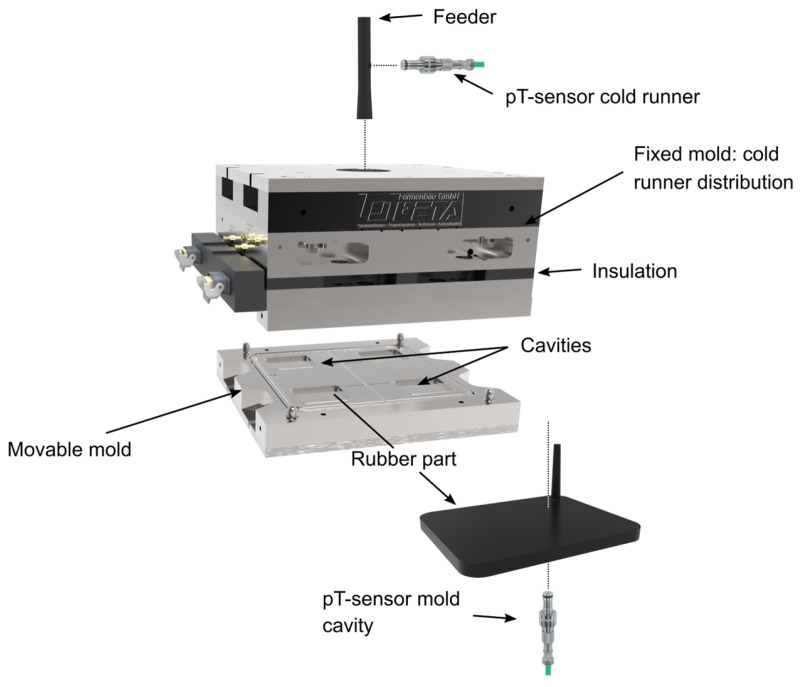
A multi-cavity-mold with a cold runner system was employed in injection molding experiments. The ultimate rubber parts display a volume of 100 × 80 × 6.3 mm^3^.

**Figure 3 polymers-16-02033-f003:**
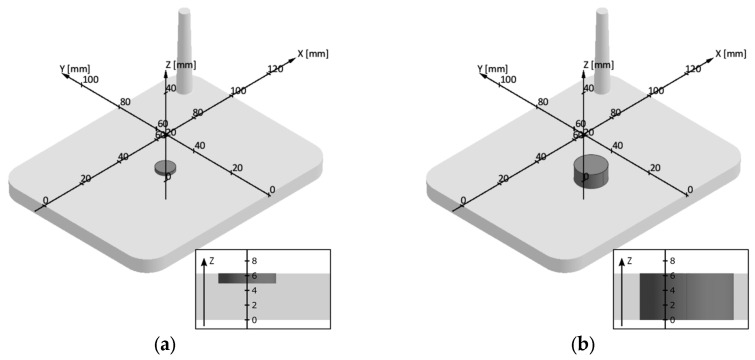
Part characterization was exhibited in central position of the cured rubber parts. (**a**) Dynamic mechanical testing followed Hutterers [6] method to obtain the dynamic spring constant upon the device equipped with a cylindrical testing probe of 8 mm in diameter in displacement-driven test mode. (**b**) Compression set analysis was conducted according to DIN-ISO 815-1 standard [23], with samples being compressed by 25% and stored at 70 °C for 24 h.

**Figure 4 polymers-16-02033-f004:**
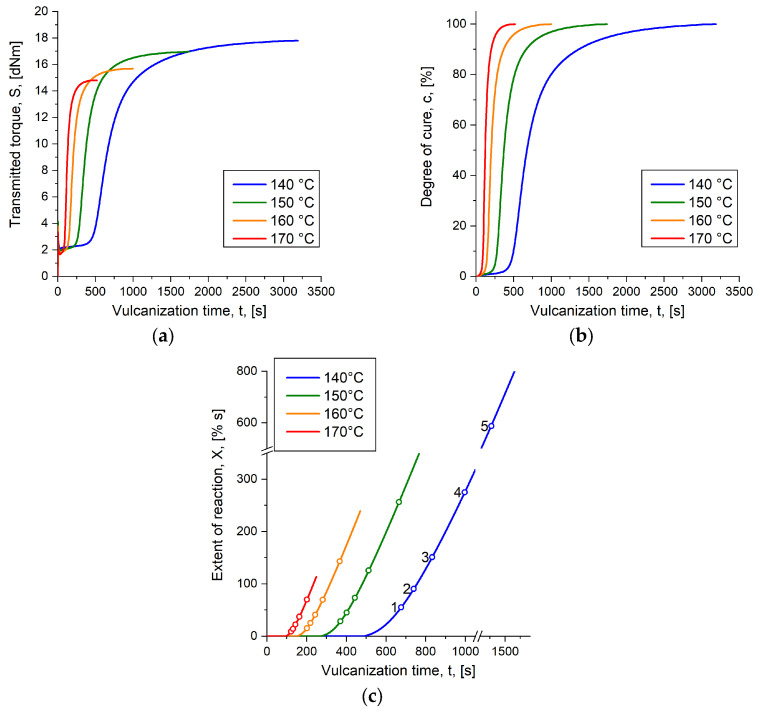
Reaction kinetics were derived from RPA measurements. The figure indicates the results being relevant for the reference: (**a**) Depicts the transmitted torque S measured at four temperatures: 140 °C, 150 °C, 160 °C and 170 °C. (**b**) Shows the consecutive degree of cure c, derived from the raw data. (**c**) Indicates the extent of reaction X, which is determined according to Equation (6). The circles mark the extent value at a specific degree of cure, whereby the first one (marked as 1) describes X at c = 50% and the last one (marked as 5) corresponds to X at c = 90%.

**Figure 5 polymers-16-02033-f005:**
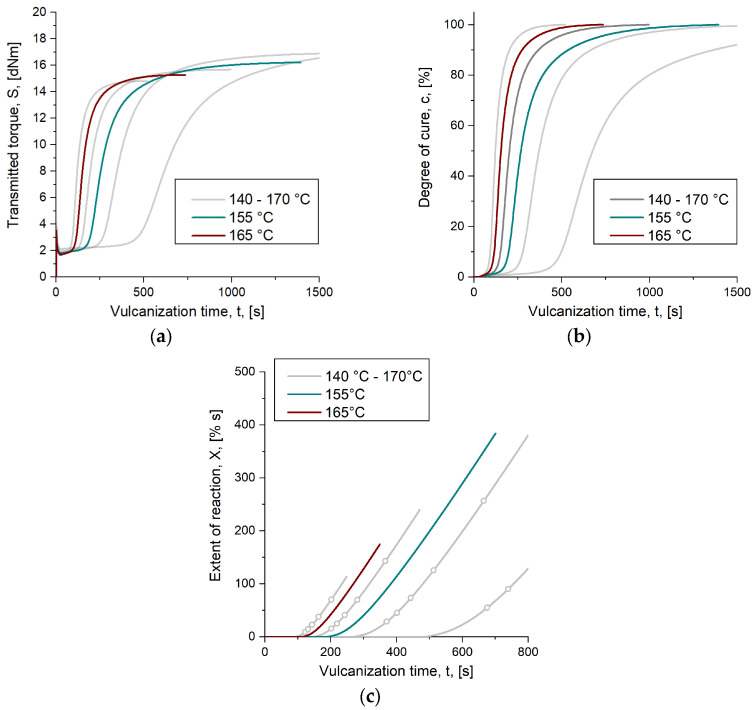
Reaction kinetics were derived from RPA measurements. The figure indicates the results being relevant for the validation experiments: (**a**) represents S measured at 155 °C and 165 °C, which were the vulcanization temperatures in the model validation experiments; (**b**) marks the normalized reaction kinetics valid for the approval; (**c**) outlines the progress of X at the given temperatures.

**Figure 6 polymers-16-02033-f006:**
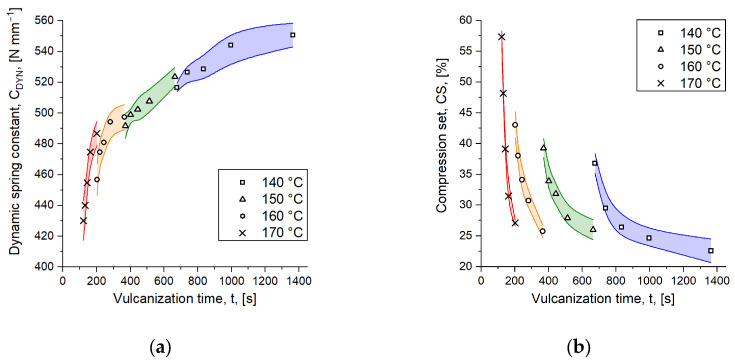
Mechanical characteristics of injection-molded rubber parts obtained from the initial reference phase, where parts were produced at the given temperatures, considering vulcanization times according to the degree of cure in the range of c = 50–90%. Dynamic mechanical analysis was employed to determine (**a**) the dynamic spring constant C_DYN_. In addition, (**b**) the compression set was assessed. The symbols mark the calculated mean value out of five measurements per time step at the vulcanization temperatures given. By the colored areas, the natural deviation of the test results is indicated per temperature.

**Figure 7 polymers-16-02033-f007:**
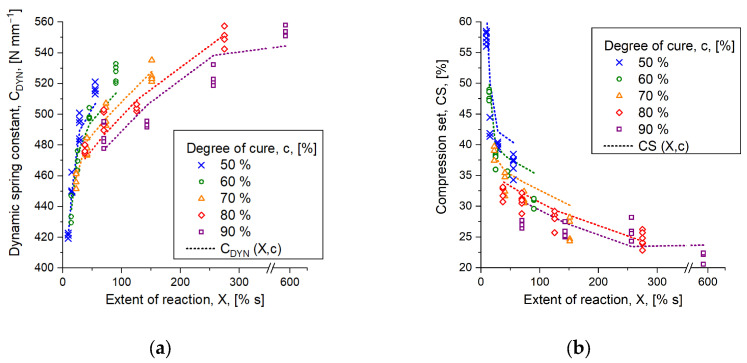
Resulting approximation curves of the combined generalized logistic growth and second order model, fitting the mechanical characteristics of injection-molded rubber parts observed upon curing to the degree of cure given at temperatures between 140 and 170 °C. Data are plotted as (**a**) dynamic spring constant C_DYN_ and (**b**) compression set CS over the extent of reaction X.

**Figure 8 polymers-16-02033-f008:**
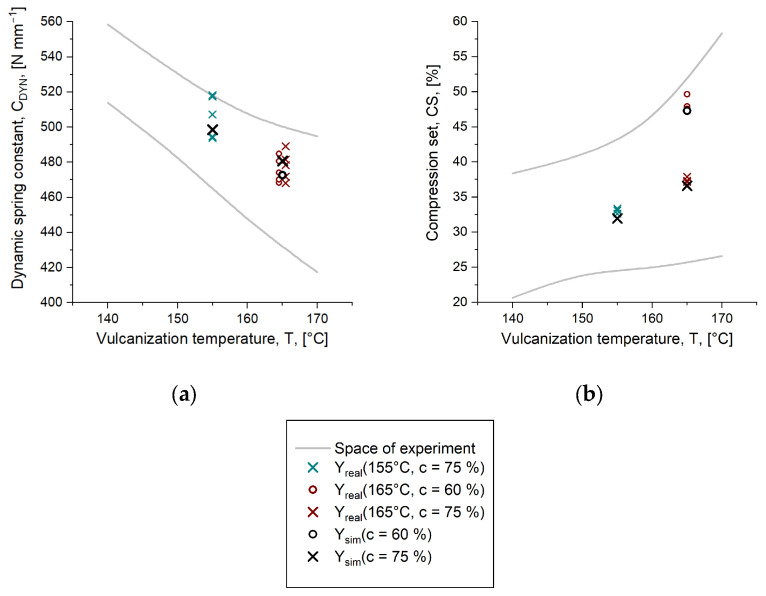
Validation results aiming to indicate the applicability of the herein proposed approach to determine mechanical part characteristics via injection molding simulation. The predictions, represented by the black symbols, are found well within the natural deviation of the real parts for each of the considered process settings. The space of experiment is derived from reference data obtained from parts cured to the lowest (c = 50%) and the highest (c = 90%) degree of cure. Data are plotted as (**a**) dynamic spring constant C_DYN_ and (**b**) compression set CS over the temperature T.

**Table 1 polymers-16-02033-t001:** Process settings for producing injection-molded rubber parts at four different vulcanization temperatures. The vulcanization times were defined for identical degrees of cure at each temperature. Apart from the time, constant process settings were applied in the reference phase realization.

Reference Data						
	T_VULC_	[°C]	140	150	160	170
Prepare mold (Mold closed)	Heating time	[min]	>90			
	Temperature feeder	[°C]	60			
	Temperature plasticizing cylinder	[°C]	70			
	Temperature screw ante chamber	[°C]	80			
	Temperature cold runner	[°C]	80			
Preparation (Mold closed)	Screw speed	[min^−1^]	90			
	Dosing volume	[cm^3^]	208			
	Back pressure	[MPa]	0.2			
Injection (Mold closed)	Injection volume flow rate	[cm^3^ s^−1^]	24.3			
	Pressure limitation	[MPa]	22			
Vulcanization (Mold closed)	t_VULC1_ (c = 50%)	[s]	676.2	370.8	202.2	122.4
	t_VULC2_ (c = 60%)	[s]	739.8	401.4	219.0	131.4
	t_VULC3_ (c = 70%)	[s]	833.4	444.0	243.0	144.0
	t_VULC4_ (c = 80%)	[s]	996.6	513.0	281.4	163.8
	t_VULC5_ (c = 90%)	[s]	1360.8	665.4	366.6	202.2
Retention time	Mold opening	[s]	50–60			
	Demolding/Cleaning					
	Mold closing					
Cooling	Time	[min]	>10			

**Table 2 polymers-16-02033-t002:** Process settings applied in the experimental validation of the proposed model approach to predict mechanical characteristics of parts manufactured from the SBR compound. Pressure settings are given as hydraulic pressure.

Validation Data					
	T_VULC_	[°C]	155	165	
Prepare mold (Mold closed)	Table 1				
Preparation (Mold closed)	Screw speed	[min^−1^]	90		90
	Dosing volume	[cm^3^]	208		
	Back pressure	[MPa]	0.2		
Injection (Mold closed)	Injection volume flow rate	[cm^3^ s^−1^]	24.3		24.3
	Pressure limitation	[MPa]	22		
Vulcanization (Mold closed)	t_VULC1_	[s]	---	167.4	
	t_VULC2_	[s]	355.8	196.8	
Retention time	Mold opening	[s]	~50–60		
	Demolding/Cleaning				
	Mold closing				
Cooling	Time	[min]	>10		

**Table 3 polymers-16-02033-t003:** Overview of process-relevant settings for simulating the validation experiments conducted on the injection molding machine. SIGMASOFT^®^ Version 6.0 was employed for the calculations. The ability of considering a time-dependent profile of the actual state of the mass temperature for the dosing phase is declared in the appendix. The initial temperature is derived from this profile. Pressure-related conditions are default values suggested by the software.

Simulation Data				
	T_VULC_	[°C]	155	165
Prepare mold (Mold closed)	Heating-Up	[-]	PI-controller automatic	
Dosing phase	Initial temperature	[°C]	93.0	93.0
	Temperature profile	[-]	Appendix	
Injection phase	Filling time	[s]	8.9	8.9
	Max. inlet pressure	[MPa]	22	
	Max. inlet pressure increase rate	[MPa s^−1^]	1500	
	Max. inlet pressure decrease rate	[MPa s^−1^]	80	
Vulcanization phase	t_VULC1_	[s]	---	167.4
	t_VULC2_	[s]	355.8	196.8

**Table 4 polymers-16-02033-t004:** The extent of reaction X is derived from the vulcanization isotherms obtained from the RPA measurements by applying Equation (6), aiming to determine the area beneath the curve in the range of incubation time and the vulcanization time required to approach a specific degree of cure c. The results shown include X for the reference data, which was employed in the calculation of the approximation coefficients, and X of the later-discussed validation experiments.

Extent of Reaction *X* [% s]					
	Degree of Cure	T = 140 °C	T = 150 °C	T = 160 °C	T = 170 °C
Reference data	c = 50%	55.1	28.6	15.5	9.2
	c = 60%	90.3	45.2	25.2	14.2
	c = 70%	151.2	73.4	40.9	22.5
	c = 80%	275.0	125.6	69.5	37.5
	c = 90%	587.3	256.5	143.2	70.0
	Degree of cure	T = 155 °C	T = 165 °C		
Validation data	c = 60%	---	19.3		
	c = 75%	79.2	39.5		

**Table 5 polymers-16-02033-t005:** Fit coefficients of the generalized logistic growth function for predicting the mechanical characteristics of injection-molded rubber parts from process simulations. The coefficients were derived from experimental results obtained upon dynamic mechanical analysis and compression set analysis. The predictability of mechanical part characteristics by the two functions is given by the coefficient of determination R^2^.

Generalized Logistic Growth Function Coefficients									
	K	α_1_	α_2_	β_1_	β_2_	γ	δ	ε	R^2^
C_DYN_	519.37	−0.7111	−0.0015	0.5247	−0.0006	0.2392	23.3455	24.7962	0.9251
CS	67.50	−0.6189	0.0029	−0.0649	0.0001	0.2902	13.0072	4.4565	0.8596

**Table 6 polymers-16-02033-t006:** Summary of mechanical characteristics measured from real parts and obtained upon simulation prediction. The values dedicated to reality are given as median in combination with the absolute deviation to minimum and maximum. Simulation results do not deliver any kind of statistical value since the material data sets were kept constant throughout all simulations.

Validation Results							
T_VULC_	[°C]	155		165		165	
c	[%]	75		60		75	
C_DYN, REAL_	[N mm^−1^]	507.2	_+11.2_	474.0	_+10.8_	478.3	_+10.8_
			^−13.5^		^−5.8^		^−10.3^
C_DYN, SIM_	[N mm^−1^]	498.6		472.5		480.7	
CS_REAL_	[%]	32.6	_+0.7_	47.6	_+2.1_	37.3	_+0.6_
			^−0.7^		^−0.4^		^−0.6^
CS_SIM_	[%]	31.9		47.3		36.6	

## Data Availability

The data presented in this study are available upon request from the corresponding author and after consultation with our partners. The data are not publicly available as they are part of an ongoing study.

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
