# Peer review of "Optimization Strategy for Process Design in Rubber Injection Molding: A Simulation-Based Approach Allowing for the Prediction of Mechanical Properties of Vulcanizates"

_polymers, 2024, doi:10.3390/polym16142033_

Round 1
Reviewer 1 Report
Comments and Suggestions for Authors
The manuscript by Traintinger et al. proposes a mathematical model for optimization of process setup in rubber injection molding. While the subject of the paper is interesting, I do not see enough innovation and progress in their proposed model. The main finding of this work is Equation 5, which is a combination of two existing models, Equation 4 (for prediction of sigmoidal behaviour of kinetic measurements in rubber processing) and Equation 1 (for prediction of mechanical properties), respectively. The proposed Equation 5, consists of eight fitting parameters, which obviously should work better than Equation 4, consisting of four adjustable parameters. Meanwhile, it is not completely clear (at least not justified) how the authors combine Equations 1 and 4; they replace the parameter F (upper asymptote) in Equation 4 with a second-order polynomial for Y (which is not the upper asymptote, rather it is a time-dependent quantity). Moreover, Equation 5 does not seem to be correct form the point of view of dimensions of its parameters; the parameter X (extent of reaction), defined in Equation 6, has the dimension of c times t (at least it does not have the same dimension as c). Another point is that, while the proposed Equation 5 is based on the extension of Equation 4, in the results no comparison between predictions of the two models is made. This is essential, to see how a combination of Equations 1 and 4 improves the prediction ability of the proposed model. For these reasons, I cannot recommend publication of this paper.
Reviewer 2 Report
Comments and Suggestions for Authors
1. This is certainly a good paper which has an interesting technology content , but significant improvement can be reached from the scientific viewpoint . Some suggestions are reported in the points below.
2. The abstract is more than 75% devoted to describe the motivation , whereas the real research work done is limited to the last few lines. IT should be rewritten limiting motivation to few lines and devoting more room to information about the innovative research described in the ms.
3. The section ; 2. Theoretical background even if not my field of expertise, looks like a review of calculations described in the literature. It is very long and not essential for the average reader. It could be moved in the g added information out of the main ms.
4. Materials: the authors report that they are using a “SBR compounds” , but “De tailed quantities of the compound’s ingredients cannot be disclosed due to its application for industrial purposes”.
In my firm opinion is not easy to accept to calculate and measure properties of a compound without declaring the detailed COMPOSITION. Different species structure and content are of course determining the behavior and the final properties. This information lacking all the data loose their utility for the scientific community.From a scientific view point I should stop my review at this point, however I complete my report for the authors as the contribution has a significant technology interest and in the hope they can overwhelm the above problem
5. The extent of reaction used in the description of the results ,if I understand well is ,calculated by considering 100% the value reached at the stationary state of the vulcanisate properties. This is certainly a valid empirical approach but the shape of curves and their time dependence , certainly depends on the compound detailed composition. This is well understood by the authors in the final part of the conclusion reporting that with peroxides as crosslinking agent the situation is different!
6. It is my conclusion that the paper is for sure a very good contribution to technical knowledge for the prediction of the complex vulcanization process and final properties. But the scientific value and more generalized utility are strongly limited by the lack of detailed chemical information on the system investigated.
7. The paper is interesting but very long and many well known detail are reported. Sections 3 and 4 should be rewritten in a more synthetic manner by removing well known information-
Comments on the Quality of English Language
Just a moderate revision
Round 2
Reviewer 1 Report
Comments and Suggestions for Authors
The authors have revised the manuscript according to my comments.
Reviewer 2 Report
Comments and Suggestions for Authors
I understand the problem of getting composition of commercial polymer formulation ; the authors looks to be aware
However the method they propose is valid and the revised paper can be on my side accepted for publication